

# 1 Exploring the potential relationship between the occurrence
# 2 of landslides and debris flows: A new approach

**Zhu Liang[1],Changming Wang[1] and Kaleem Ullah Jan Khan[1]**
(College of Construction Engineering, Jilin University, 130000 Changchun, People's Republic of
China)
E-mail:wangcm@jlu.edu.cn
**Abstract:** The aim of the present study is to explore the potential relationship between landslides
and debris flows by establishing susceptibility zoning maps separately with the use of random
forest. Longzi township, Longzi County, located in Southeastern Tibet, where historical landslide
and debris flow are commonly occurred, was selected as the study area. The work has been carried
out with the following steps: **(1)** A complete landslide and debris flow inventory map was
prepared; **(2)** Slope units and 11 controlling factors were prepared for the susceptibility modelling
of landslide while watershed units and 12 factors for debris flow; **(3)** Establishing susceptibility
zoning maps for landslide and debris flow, respectively, with the use of random forest; **(4)** The
performance of two models are verified using ROC curve, the values of AUC and contingency
tables; **(5)** Putting the high or very-high-class watershed units in the debris flow susceptibility
zone map as the base map to observe its coverage by slope units of different classes; **(6)** The
landslide zoning map was put at the bottom floor and analyzed the distribution of high or
very-high-class slope units in watershed units; **(7)** transforming the slope units into points and
distributed them on the watershed units. Two models based on random forest have demonstrated



great predictive capabilities, of which accuracy was close to 90% and the AUC value was close to
1. The loose sources carried out by the debris flows are not necessarily brought by the landslides
although most landslides can be converted into debris flows. The area prone to debris flow does
not promote the occurrence of landslides. A susceptibility zoning map composed of two or more
natural disasters is comprehensive and significant in this regard.
**Key words:** Landslide; Debris flow; Susceptibility; Random forest; Potential relationship

# 29  1. Introduction

Landslides and debris flows are natural phenomenon mainly occurring in mountainous areas,
which pose considerable threats to people, industries, and the environment directly or indirectly.
Generally, damages can be decreased to a certain extent by predicting the likely location of future
disasters (Pradhan, 2010). Thus, extensive research has been conducted for the prediction and
susceptibility assessment of landslides and debris flows.
In geomorphology, a "landslide" is the movement of a mass of rock, debris or earth down a
slope, under the influence of gravity (Cruden and Varnes, 1996). Debris flow is a specific type of
landslide, which can be defined as (Hungr et al. 2013): ''Very rapid to extremely rapid surging
flow of saturated debris in a steep channel''. Generally, a landslide that occurs on a steep slope and
becomes disaggregated as it tumbles down can transform into a debris flow if it contains sufficient
water for saturation. Therefore, landslide provides sufficient material source for the occurrence of
debris flow and most of the landslides were accompanied by debris flow. In the past, few scholars
have not been specifically distinguished the landslide and debris flow in terms of susceptibility





evaluation (Alessandro et al., 2015; Guzzetti et al., 2005). In addition, some scholars made
separate evaluations of landslide and debris flow (Park et al., 2011; Haydar et al., 2016). Some
scholars have proposed a coupled model of landslide-debris flow (Chiang et al., 2012; Gomes et
al., 2013). However, not every landslide has evolved into a debris flow and the material source of
the debris flow is may not a landslide. The causes and manifestations of landslides and debris
flows are different. In a debris flow, it is possible to distinguish initiation (source area), transport
and deposition zone. In other words, there is no necessary connection between debris flow and
landslides. Besides, the conditioning factors and mapping units involved in the susceptibility
assessment of debris flow and landslide are not identical. Therefore, it is more reasonable to
evaluate the susceptibility of landslide and debris flow separately. As an example, a landslide
inventory map includes only landslides, as does debris flow.

The methods of susceptibility assessment can be broadly classified as qualitative or

quantitative(Aleotti et al., 1999). Several methods and approaches have been proposed and tested
to ascertain susceptibility, such as physical-based approaches (Carrara et al., 2008), heuristic
methods (Blais et al., 2016) and statistically-based approaches (Reichenbach et al., 2018). In
addition, new machine learning models, such as neural networks (Park et al.,2013), support vector
machines (Colkesen et al.,2016) and random forest (RF) ( Liu et al., 2018), have also been
applied.

The Longzi County in Southeastern Tibet is always exposed to landslide and debris flow

hazard because of climatic and topographic conditions, which is chosen as the study area. The
purpose of the present study is to explore the potential relationship between the occurrence of
landslides and debris flows by establishing susceptibility zoning maps separately with the use of



random forest.

## 2. Materials

### 2.1  Study area

The study area located in Longzi Township, Longzi County, Southeastern Tibet is bounded by
longitudes of 92°15'E and 92°45'E, latitudes of 28°10'N and 28°30'N (Fig.1). It covers an area of
about 535 km$^2$ with a population of more than 6000. The study area belongs to a semi-arid
temperate monsoon climate with the annual rainfall of 279 mm, mainly concentrated in May to
September. The seismic intensity within the area has a degree of VIII on the modified Mercalli
index.

The study area belongs to the zone of stratigraphic division of the Northern Himalayan block.

The strata is mainly composed of Mesozoic Cretaceous, Jurassic, Triassic, and Cenozoic units.
There were three common lithology observed during our field investigation: Siltstone from the
Laka Formation ($K_1l$); Conglomerates from the Weimei Formation ($J_3w$) and Quaternary slope
wash ($Q_4^{el+dl}$) from the Cenozoic strata.

The disasters in the study area mainly consist of rain-fed high frequency debris flows and

landslides, which destroyed and flooded roads, bridges, farmlands, villages, etc., causing great
economic losses.

### 2.2  Landslide and debris flow inventory

The statistically-based susceptibility models are based on an important assumption: future
landslides will be more likely to occur under the conditions which led to the landslides past and





present (Varnes, 1984; Furlani and Ninfo, 2015). Therefore, a complete and accurate inventory
map is the key for model training and validation. In this study, data comes from historical records,
field surveys (**Fig.2 and Fig.3**) and interpretation of Google Earth images carried out in Google
Earth pro 7.1(**Fig.4**). Finally, a total of 396 landslide points and 49 debris flow points were
recorded and mapped (**Fig.1**).

## 90    2.3  Mapping units

The selection of the mapping unit is an important pre-requisite for susceptibility modelling
(Guzzetti, 2006). The main mapping units commonly used for landslide and debris flow
susceptibility assessment are grid cells (Reichenbach et al., 2018). Despite its popularity and
operational advantages, grid-cells have clear drawbacks for susceptibility modelling (Guzzetti et
al., 1999). There is no physical relationship between a grid-cell, while slope units can make up for
this deficiency. Depending on the landslide type, a slope unit may correspond to an individual
slope, an ensemble of adjacent slopes or a small catchment (Reichenbach et al., 2018). The
geometry of debris flow is better represented by apolygon or a set of polygons in vector format. In
the present study, adjacent slope units were applied to the susceptibility assessment of landslides.
First-order sub-catchments, which is also called watershed unit, was applied to the susceptibility
of debris flow (Francesco et al., 2015; Qin et al.,2018). Therefore, ArcGIS is used in this paper to
divide the study area into 174 catchments or 1003 slope units and make artificial corrections
according to remote sensing image.

## 104   2.4  Controlling factors and mapping

The selection of evaluation parameters is another key prerequisite to ensure that the model is


accurate and reasonable. With reference to previous studies (Ahmed et al., 2016; Xu et al., 2013;
Braun et al., 2018), there are differences in the controlling parameters used in landslide and debris
flow susceptibility assessment. The occurrence of debris flow emphasizes the indispensability of
provenance, topography and triggering factors. Availability, reliability, and practicality of the
factor data were also considered (van Westen et al., 2008). In this paper, 11 landslide controlling
factors are selected, including distance to fault, distance to road, distance to river, annual rainfall,
slope angle, aspect, plan curvature, profile curvature, topographic wetness index, elevation and
maximum elevation difference. Besides, a total of 12 controlling factors, including basin area,
main channel length normalized difference vegetation index (NDVI), drainage density, roundness,
melton, average gradient of main channel, slope angle, maximum elevation difference, annual
rainfall, distance to fault and elevation were selected to fully reflect the characteristics of the
watershed for the susceptibility assessment of debris flow.
The controlling factors in the present study can be categorized into four types: **(1)** The
morphological factors (slope, aspect, plan curvature, profile curvature, roundness, melton); **(2)**
Geological factors (distance to fault, basin area, main channel length, drainage density); **(3)**
Topographical factors (elevation, maximum elevation difference, average gradient of main
channel ); **(4)** Environmental factors (annual rainfall, topographic wetness index, NDVI, distance
to road, distance to river). Totally 18 factors are obtained by processing the row data in the ArcGIS
10.2 platform. Morpholigical and topographic related factors were derived from the DEM with a
resolution of 30 × 30 m. Geological related factors were extracted from 1:50000 geological maps.
Rainfall is one of the most important external factors inducing landslides and debris flow, which
was determined by ordinary kriging interpolation in ArcGIS by collecting data of 6 precipitation


stations near the area under study as a reference.

## 2.5 Mapping

In the current study, the maps of controlling factors were reclassified into 4 to 7 classes based on
the equal spacing principle and the mean value in the unit was counted as the representative value
of the unit. Aspect, which is frequently used as landslide controlling factor (Dai and Lee, 2002),
was reclassified into 8 classes (**Fig.2** ). Plan curvature and profile curvature were both considered
and were both reclassified into six classes. Generally, faults, rivers and roads play a key role in the
occurrence of landslides and were reclassified into seven classes using an interval of 1500m
(**Fig.2**). Topographic wetness index was reclassified into five classes (**Fig.2** ).

NDVI reflects the vegetation conditions in the area and was reclassified into 5 classes(**Fig.3**).

Drainage density is the ratio of the total drainage length to the watershed area and was reclassified
into six classes (**Fig.3**). Roundness refers to the ratio of the area of a basin to the area of a circle
with the same circumference and was reclassified into six classes (**Fig.3**) . Melton ratio refers to
the ratio of the degree of undulation in the watershed to the square root of the arithmetic area of
the watershed (Melton, 1965), which is reclassified into seven classes (**Fig.3**). Considering the
correlation between the two controlling factors, basin area and main channel length are
represented by the same graph, which was reclassified into four classes (**Fig.3**). Average gradient
of main channel , which is the ratio of the maximum elevation difference of main channel to its
linear length, was reclassified into six classes (**Fig.3**).
Rainfall is the only triggering factor to be considered for both landslide and debris flow in this paper,
which was reclassified into six classes (**Fig.2 and Fig.3**). Slope angle is frequently employed in both



landslide and debris flow susceptibility mapping and was reclassified into six classes (**Fig.2 and Fig.3**).
Maximum elevation difference reflects the kinetic energy condition and is reclassified into 6 classes
using an interval of 200m (**Fig.2 and Fig.3**). Elevation was reclassified into five classes (**Fig.2 and**
**Fig.3**), which has also been used by many authors (Ayalew and Yamagishi, 2005; Pourghasemi et al.
2013a, b ).

## 154   3. Methods

## 155   3.1 Sampling strategies and validation

Statistical models for landslide susceptibility zonation reconstruct the relationships between
dependent and independent variables using training sets, and verify these relationships using
validation sets (Guzzetti et al., 2006a,b), which usually implies the partitioning of the inventory in
subsets. The sampling strategy affects the results of the susceptibility map (Yilmaz, 2010). Based
on temporal, spatial or random criteria, the partition of landslide inventories can be made (Chung
and Fabbri, 2003) and the most applied one is a one-time random selection (Reichenbach et al., 2018).
In the current study, the random partition was used due to existing constrains with the temporal and the
spatial partition. Therefore, sample data was divided into two parts: 70% of the data was selected as
training data to create a prediction model, and the remaining 30% of the data was used for validation.

The computation of the area under the curve (AUC) is the most popular metrics to estimate

the quality of model , which has been applied for ROC curves( Green and Swets, 1966). It is one
of the most commonly used indicators. A typical two-entry confusion matrix, including true
positives (TP), true negatives (TN), false positives (FP), and false negatives (FN), is another
common index. In current study, both ROC curve and the contingency tables were used to

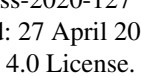



evaluate the susceptibility models established for landslides and debris flow.

## 3.2 Random Forests

Random forest (RF) is a powerful ensemble-learning method and was first introduced by Breiman
(2001). RF uses the bagging technique (bootstrap aggregation) to select, at each node of the tree,
random samples of variables and observations as the training data set for model calibration.
Unselected cases (out of bag) are used to calculate the error of the model (OOB Error). The
increase in OOB error is proportional to the importance of the predictive variable (Breiman and
Cutler 2004). There are no restrictions on the types of variables, either numerical or categorical.
RF has the ability to reduce errors caused by unbalanced data, which is suitable for susceptibility
assessment.
In this study, the scikit-learn package (Pedregosa et al.,2011) in the programming software
python version 3.7 was used for the modeling. The number of trees (k) and the number of
predictive variables used to split the nodes (m) are two user-defined parameters required to grow a
random forest (Ahmed et al.,2016). In order to ensure the algorithm convergence and good
prediction results, the number of trees (k) has been fixed to 500 and the number of predictive
variables (m) has been selected as 5 (Breiman et al.,2001).

## 4. Results and verification

## 4.1  Landslides susceptibility mapping results

In this study, the predictive accuracy, ROC curves and AUC values of the RF model using training
data are showed in **Table 1** and **Fig. 4**. The RF model ensured very high TN and TP values of


92.86% and 93.57%, respectively. An AUC equals to 1 indicates perfect prediction accuracy
(Vorpahl et al., 2012). The RF model has great performance in terms of AUC, with value of 0.978.
Standard error (St.), confidence interval (CI) at 95% and significance (Sig.) are applied as three
evaluation statistics. All these results indicate a reasonable goodness-of-fit for models with the
training dataset, for which the values are reasonably small.

The task of validating the predicted results is the critical strategy in prediction models as

shown in **Table 3** and (**Fig. 4**). Consequently, the values of TN and TP were 92.90% and 90.0%,
respectively. It can be seen that the model has also a great performance in terms of AUC with
value of 0.977. In comparison with the training model, the accuracy and AUC values have slightly
decreased, but still perform well.

The landslide susceptibility map was also reclassified into five classes: very low (0~0.2), low

(0.2~0.4), moderate (0.4~0.6), high (0.6~0.8), very high (0.8~1) by using the equal spacing
method (**Fig.5** ). The maps should satisfy two spatial effective rules: (1) The existing disaster
points should belong to the high-susceptibility class and **(2)** The high-susceptibility class should
cover only small areas (Bui et al. 2012). The number of units belonging to very high class reached
179, accounting for 17% (**Fig.6**). Disaster points were mostly in the dark (red or orange) areas.
The units belonging to moderate class accounted for the smallest proportion, at 13% (**Fig.7**).

The controlling factors with significant effects were selected and normalized as shown in

**Table 2**. The weight values of slope angle, distance to fault, plan curvature and topographic wetness
index was 0.21, 0.19, 0.17, 0.13 respectively, which was closely related to the occurrence of
landslide. The weight values of distance to road, maximum elevation difference, profile curvature
and elevation are less than 0.1 as 0.08, 0.08, 0.06, and 0.05, respectively (**Fig.7**).



## 4.2 Debris flow susceptibility mapping result

The debris flow susceptibility model perform well with a very high TN and TP values as 90.90% and 91.18%, respectively. In terms of AUC, the model has also a great prediction performance with the value of 0.979 (**Fig.4**). Three evaluation statistics also indicate a reasonable goodness-of-fit for the model.

**Table 1** shows that in the 30% sample data used for verification, the values of TN and TP were 89.13% and 86.67%, which were slightly decreased compared to the training model. It can be seen that the model has also a great performance in terms of AUC, with value of 0.968.

The number of units belonging to very high-class reached to 26, which is accounting for 15% while the units belonging to high-class accounted for the smallest proportion at 13%. More than half of the units (58%) belong to on a low or very low-class (**Fig.6**). Disaster points were mostly in the dark (Bright or deep red) areas (**Fig.5**).

The weight values of main channel length, roundness and slope angel were 0.25, 0.16, 0.14 respectively, which has significant influence on the occurrence of debris flow. The weight values of elevation, maximum elevation difference, melton and basin area are close to 0.1, which are 0.13, 0.12, 0.1, and 0.1 respectively(**Fig.7**).

## 4.3 Analysis and comparison of landslide and debris flow susceptibility

It is worth comparing the two susceptibility zonation. In terms of prediction accuracy, the values of TP, TN and AUC of landslide model are slightly higher than that of debris flow. However, both models achieved high predictive performance. Therefore, the landslide and debris flow



susceptibility assessment models based on random forest are reliable. The purpose of the present
study is to explore the potential relationship between landslides and debris flows by establishing
susceptibility zoning maps. Figure 8 shows the overlapping distance between debris flow and
landslide in high or very high-class of susceptibility areas. It can be seen from the figure that most
of the areas with high or very high-class in the map of debris flow are covered with landslides.
However, there are also non-overlapping areas between the two zoning maps. There are 23 units
belonging to high-class in the debris flow susceptibility zoning map (**Fig.8**), of which 17 units are
covered with high or very high-class units in the landslide zoning map (**Table 4**). In addition, there
are 4 watershed units covered with low or very low class slope units. In the same way, 19
watershed units belonging to very high-class are covered with high or very high-class slop units
and 4 watershed units with low or very low-class slop units. In other words, more than 70% of the
high or very high-class watershed units are covered with high or very high-class slope units.
However, there are still 30% of watershed units with high or very high-class without the
distribution of slope units in corresponding grades. It validated the previous view that most of
landslides can be transformed into debris flows. Factor analysis was applied to further analyze the
reasons for the difference. 36 watershed units with distribution of high-grade slope units were
taken as model 1 and the left 8 watershed units as model 2. The KMO (Kaiser-Meyer-Olkin)
statistic test values were 0.766 and 0.643 respectively, which indicated that the correlation
between variables is obvious and suitable for factor analysis (**Table 5**). In model 1, the cumulative
contribution rate of the three factors (C1, C2 ,C3 ) reached to 83.6%, while the cumulative
contribution rate of the first four factors (F1, F2 ,F3 and F4 ) reached to 80.5% for model 2 (**Table**
**6**). According to the correlation coefficient of each common factor (**Table 6**), the first common



factor mainly highlighted the information of basin area, main channel length and maximum
elevation difference. Similarly, the second and the third common factor highlighted the
information of slope angle and elevation and roundness, respectively. The difference between the
two models is that the second model has the fourth common factor (**Table 7**), which emphasized
the effects of rainfall and distance to the fault. The transformation from a landslide to a debris
flow most often occurs during heavy rainfall (Takahashi, 1978), and the landslides are the source
area. But landslides are not the only source of debris flows. The loose material distributed in the
basin is not necessarily caused by landslide.

In turn, we analyze the distribution of high or very high-class slope units in watershed units.

The landslide zoning map was put at the bottom floor and the debris flow zoning map on the top
floor (**Fig8**). There are 167 slope units belonging to high-class in the landslide susceptibility
zoning map (**Fig.6**), of which 68 units (accounting for about 40%) are distributed in the area of
high or very high-class watershed units in the debris flow zoning map (**Table 8**). Besides, 69 slope
units (accounting for about 41%) are distributed in the area of low or very low-class watershed
units. Similarly, 53 slope units (accounting for about 30%) belonging to very high-class are
distributed in the area of high or very high-class watershed units and 88 slope units (accounting
for about 50%) in low or very low-class slop units (**Table 8**). Comparing with the extent of the
landslide affecting the debris flow, the impact of the debris flow on the landslide is not obvious.
This indicates that the area prone to debris flow does not promote the occurrence of landslides.

Finally, we took the center of gravity of 1,003 slope units as the potential hazard points and

spread them over 174 watershed units. Thus, a combining susceptibility prediction map for
landslide and debris flow was obtained (Fig.8). The darker the color, the higher the class of



susceptibility will be. It can be seen from the figure that the level of disaster susceptibility in the
south is generally higher than that in the north, and the area in the southwest is disaster-prone. The
northeast and central locations in the area are less likely to be affected by disasters and belong to
low-susceptibility areas. Green or yellow dots, which refer to slope units with very low or low-
class in the landslide zoning map, mainly distributed in light-colored areas but there are also quite
a few green or yellow dots distributed in dark areas, which means that the occurrence of debris
flow not necessarily depend on landslides. Blue or black spots are mainly distributed in dark areas
but there are also quite a few blue or black spots distributed in dark light areas, which means that
landslide is not the only condition for debris flow to occur. Most of the watershed units are
distributed with two or more colored dots, which means that there would be multiple slope units
with different susceptibility class in the same watershed. According to the susceptibility zoning
maps of landslide and debris flow, the study area can be divided into 4 categories: **(1)** Low or very
low-class watershed units coupled with low or very low-class slope units; **(2)** Low or very
low-class watershed units coupled with high or very high-class slope units; **(3)** High or very
high-class watershed units coupled with low or very low-class slope units; **(4)** High or very
high-class watershed units coupled with high or very high-class slope units. We assume that the
occurrence of landslides can bring rich sources of debris flow, thereby promoting or aggravating
the outbreak of debris flow, that is, forming a landslide-debris flow disaster chain. Therefore, the
susceptibility assessment of the landslide-debris flow chain in the study area can be roughly
divided into three classes, which are low, moderate and high (**Table 8**).


## 5. Discussion

## 5.1 Method used for modeling

Many researchers have used different statistically-based methods to evaluate the susceptibility of landslides or debris flows. Logistic regression and discriminant analysis are the most popular methods to use in traditional multivariate statistical analysis. The performance of new learning machines, such as support vector machines and neural networks, has also been verified. Random forest, as a newly integrated learning machine, has less application in landslide and debris flow analysis. Actually, random forests have powerful data processing capabilities and can simultaneously solve problems such as high-dimensional, unbalanced and data loss, which are common in geological disaster assessment. Most importantly, random forests can compare the important differences between features and have strong ability to resist overfitting and generalization, which is difficult to achieve by other statistical methods.

## 5.2 Potential relationship between landslide and debris flow

There is a certain similarity in the evaluation of the susceptibility of landslides and debris flows from the concept, the selection of controlling factors and the application of modeling strategies. Therefore, some researchers have neglected the difference between landslide and debris flow i.e to express two different disasters with the same susceptibility zoning map(Ciurleo et al., 2016; Ciurleo et al., 2017; Persichillo et al., 2017;). However, similarity does not always mean consistency. Many researchers have previously conducted studies into the debris flow mobilization from shallow landslide using a coupled methodology. They are interested in the dynamic simulation of debris flow based on the prediction of landslide susceptibility(Wang et al., 2013; Fan



et al., 2017). However, not every landslide evolves into a debris flow, which means that the
analysis process is highly selective or uncertain. In the same way, the source of the debris flow is
not limited to landslides. Therefore, the potential relationship between landslides and debris flows
needs to be discussed more reasonably and effectively. There, the potential relationship between
landslides and debris flows needs to be discussed more reasonably and effectively. In this paper,
the corresponding influencing factors and mapping units are selected to establish landslide and
debris flow susceptibility zoning maps, respectively. The potential relationship between landslide
and debris flow is explored in two ways: **1)** Superimposing the high or very high-class
susceptibility areas in the two maps; **2)** Transforming the slope units into points and distributed
them on the watershed units. The relationship between landslide and debris flow is illustrated by
the distribution of slope units of different grades on the watershed units with different prone
grades.

## 5.3 Necessity and feasibility of combining multiple natural disaster susceptibility zoning maps

Previous studies on susceptibility zoning mapping of disaster have agreed that one disaster
corresponds to one map. Multiple disasters may be bred simultaneously in a watershed unit and it
will cause some confusion in practical. For example, the probability of a disaster occurring in a
watershed is negligible, while the probability of another disaster occurring is high. If so, we need
to combine multiple zoning maps at the same time to give a comprehensive evaluation, which is
arduous to achieve. On the one hand, the prediction accuracy and error of different zoning maps
should be similar or even consistent. On the other hand, the dimensions of the mapping unit



should be consistent or complementary. The fact that the appropriate prediction method and
mapping units applied to the two disasters makes it possible to merge the two zoning maps .In
addition, two natural disasters with potential relationship are simultaneously reflected in the same
susceptibility zoning map, which can better guide the implementation of engineering, such as
landslide-debris flow disaster chain.

## 6. Conclusion

In this paper, susceptibility prediction models for landslides and debris flows are established
through random forest, respectively and the performance of the models are excellent in terms of
accuracy and goodness of fit. The potential relationship between landslide and debris flow is
discussed by the superimposition of two zoning maps and the following conclusions can be drawn:
(1) The landslide and debris flow susceptibility prediction models based on random forest have
great performance of accuracy and goodness-of-fit and have the ability to analyze the relative
importance of different impact factors, which is suitable for the evaluation of natural disasters;
(2) Although most landslides will be converted into debris flows, the landslides are not
necessarily the source of debris flows, and the loose sources carried by the debris flow are not
necessarily brought by the landslides;
(3) By comparing the extent of the landslide affecting the debris flow, the impact of the debris
flow on the landslide is not obvious, which indicates that the area prone to debris flow does not
promote the occurrence of landslides;
(4) A susceptibility zoning map composed of two or more natural disasters is more
comprehensive and significant, which provides valuable reference for researchers and engineering



applications.

## Data availability

The data used to support the findings of this study are included within the article.

## Author contribution:

Zhu Liang was responsible for the writing and graphic production of the manuscript. Changming Wang
was responsible for the revision of the manuscript. Kaleem Ullah Jan Khan was responsible for the
translation.

## Competing interests:

The authors declare that they have no conflict of interest.

## Acknowledgements

This work was supported by the National Natural Science Foundation of China (Grant No.

4197020250).

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





**Table 1** The prediction accuracy of RF

| Test group | 70% | | | 30% | | | 100% | | |
|---|---|---|---|---|---|---|---|---|---|
| | Total | TN | TP | Total | TN | TP | Total | TN | TP |
| Landslide (%) | 93.14 | 92.86 | 93.57 | 91.75 | 92.90 | 90.00 | 92.72 | 92.87 | 92.50 |
| Debris flow (%) | 90.98 | 90.91 | 91.18 | 88.46 | 89.19 | 86.67 | 89.08 | 88.80 | 89.80 |

**Table 2** Controlling factors assigned by the RF

| Test group | Slope angle | Distance to fault | Plan curvature | Topographic wetness index | Distance to road | Maximum elevation difference | Profile curvature | Elevation |
|---|---|---|---|---|---|---|---|---|
| Landslide | 0.21 | 0.19 | 0.17 | 0.13 | 0.08 | 0.07 | 0.06 | 0.05 |

**Table 3** Controlling factors assigned by the RF

| Test group | Main channel length | Roundness | Slope angle | Elevation | Maximum elevation difference | Melton | Basin area |
|---|---|---|---|---|---|---|---|
| Debris flow | 0.25 | 0.16 | 0.14 | 0.13 | 0.12 | 0.1 | 0.1 |

**Table 4** The overlap number of debris flow and landslide height and very high-class mapping units

| Debris flow \ Landslide | Very low | Low | High | Very high |
|---|---|---|---|---|
| High | 3/23 | 1/23 | 5/23 | 12/23 |
| Very high | 2/26 | 2/26 | 8/26 | 11/26 |





**Table 5** Statistical variables of the two models

| Model | Model 1 | Mode 2 |
|---|---|---|
| Statistical variables | | |
| KMO | 0.766 | 0.643 |
| Sig. | 0.001 | 0.003 |

**Table 6** The correlation coefficients between common factors and primitive variables

| Factor | F1 | F2 | F3 |
|---|---|---|---|
| NDVI | 0.386 | -0.336 | -0.621 |
| Basin area | 0.897 | -0.007 | 0.041 |
| Main channel length | 0.984 | 0.046 | -0.023 |
| Slop angle | -0.223 | 0.829 | 0.455 |
| Maximum elevation difference | 0.744 | 0.66 | 0.011 |
| Rainfall | -0.768 | 0.33 | 0.201 |
| Average gradient of main channel | -0.753 | 0.544 | 0.106 |
| Drainage density | -0.844 | 0.06 | 0.015 |
| Roundness | 0.331 | 0.14 | 0.818 |
| Elevation | 0.133 | 0.846 | 0.382 |
| Distance to fault | -0.16 | 0.211 | 0.421 |
| Melton | -0.625 | 0.737 | 0.149 |
| Contribution rate (%) | 41.2 | 24.7 | 16.7 |



| | | | |
|---|---|---|---|
| Accumulative contribution (%) | 41.2 | 65.9 | 83.6 |


**Table 7** The correlation coefficients between common factors and primitive variables

| Factor | C1 | C2 | C3 | C4 |
|---|---|---|---|---|
| NDVI | 0.042 | -0.079 | -0.279 | -0.813 |
| Basin area | 0.802 | -0.344 | 0.057 | 0.009 |
| Main channel length | 0.885 | 0.126 | -0.196 | 0.227 |
| Slop angle | 0.009 | 0.748 | 0.58 | -0.057 |
| Maximum elevation difference | 0.801 | 0.434 | -0.128 | 0.144 |
| Rainfall | 0.197 | -0.076 | -0.487 | 0.637 |
| Average gradient of main channel | -0.744 | 0.205 | 0.15 | -0.23 |
| Drainage density | -0.776 | -0.176 | -0.267 | 0.117 |
| Roundness | -0.014 | 0.022 | 0.896 | -0.002 |
| Elevation | 0.34 | 0.746 | 0.25 | 0.326 |
| Distance to fault | 0.31 | 0.289 | -0.344 | 0.757 |
| Melton | -0.182 | 0.932 | -0.192 | 0.061 |
| Contribution rate (%) | 29.2 | 20.3 | 15.2 | 15.8 |
| Accumulative contribution (%) | 29.2 | 49.5 | 64.7 | 80.5 |

**Table 8** The overlap number of landslide and debris flow height and very-high class mapping units

| Debris flow  Landslide | Very low | Low | High | Very high |
|---|---|---|---|---|





| | | | | |
|---|---|---|---|---|
| High | 36/167 | 33/167 | 25/167 | 43/167 |
| Very high | 48/179 | 40/179 | 25/179 | 28/179 |


**Table 9** Comprehensive evaluation of landslide-debris flow susceptibility

| Landslide \ Debris flow | Low or Very low | High or Very high |
|---|---|---|
| Low or Very low | Low | Moderate |
| High or Very high | Moderate | High |







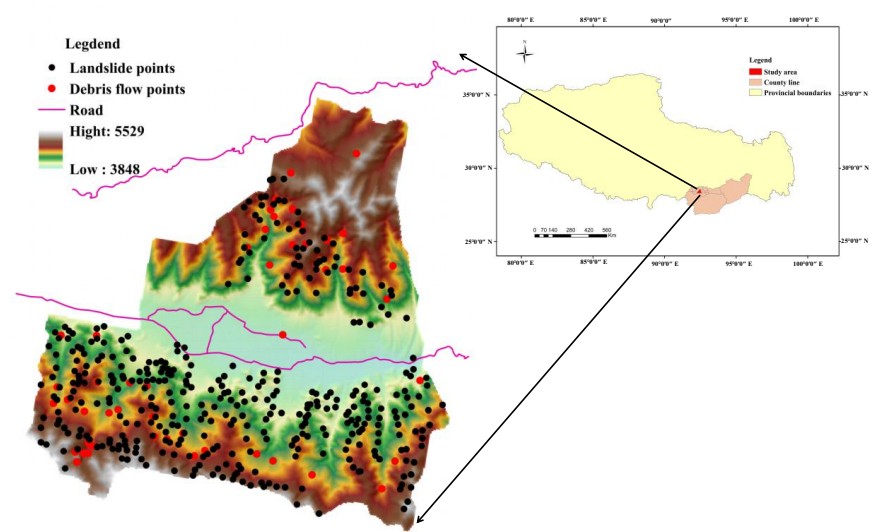


**Fig.1.** Location map of the study area showing landslide and debris flow inventory.

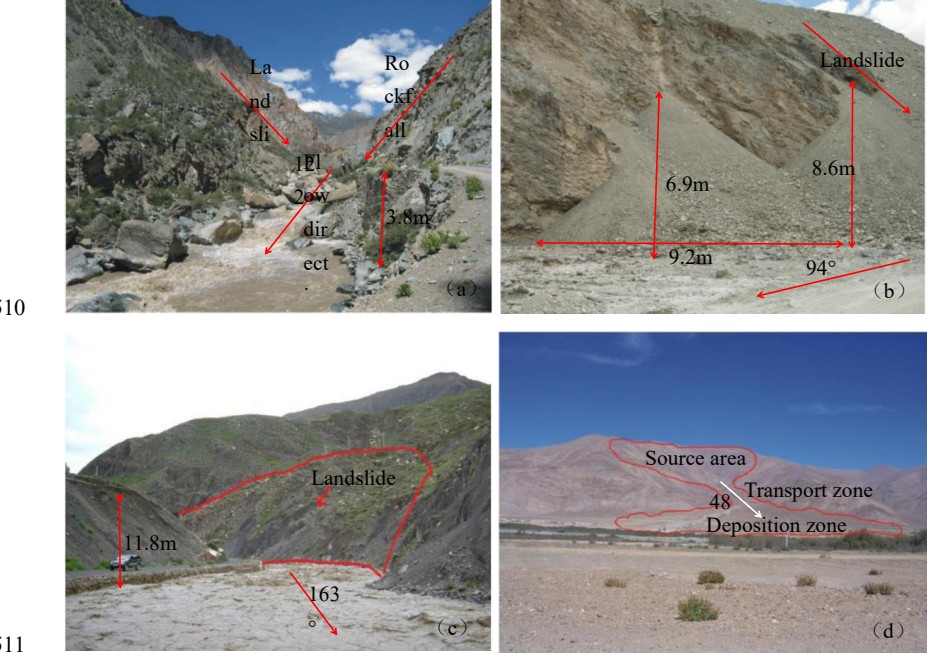



**Fig.2.** Photos of landslide or debris flow: (a) Lunba landslide in a tributary; (b) Zhenqiong landslide in
Jiayu village; (c) Debris flow in Misha Township; (d) Debris flow in Lelong Village.



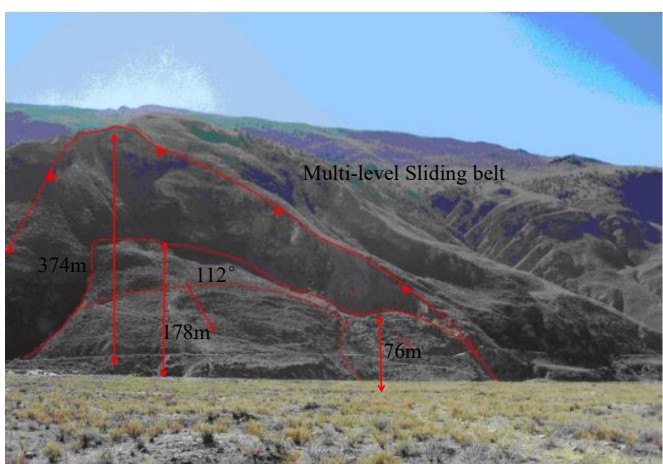


**Fig.3.** Multistage landslide in Xiongqu village

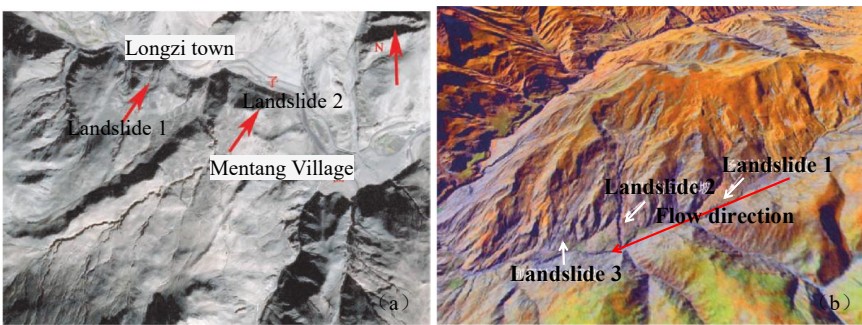


**Fig.4.** Stereo remote sensing map of landslides in Longzi Township (Tong et al., 2019)


**Fig.5.** Study area thematic maps for landslide:(**a**)Rainfall;(**b**)Profile curvature;(**c**)Maximum
elevation difference;(**d**)Average elevation;(**e**)Plan curvature;(**f**)Average slope;(**g**)Aspect;(**h**)
Wetness;(**i**)Distance to road;(**j**)Distance to river;(**k**)Distance to fault.



**Fig.6.** Study area thematic maps for debris flow: (**a**)Melton;(**b**)NDVI;(**c**)Rainfall;(**d**)Roundness;
(**e**)Maximum elevation difference;(**f**)Average elevation;(**g**)Drainage density;(**h**)Area;(**i**)
Average slope;(**j**)Average gradient of main channel;(**k**)Distance to fault.

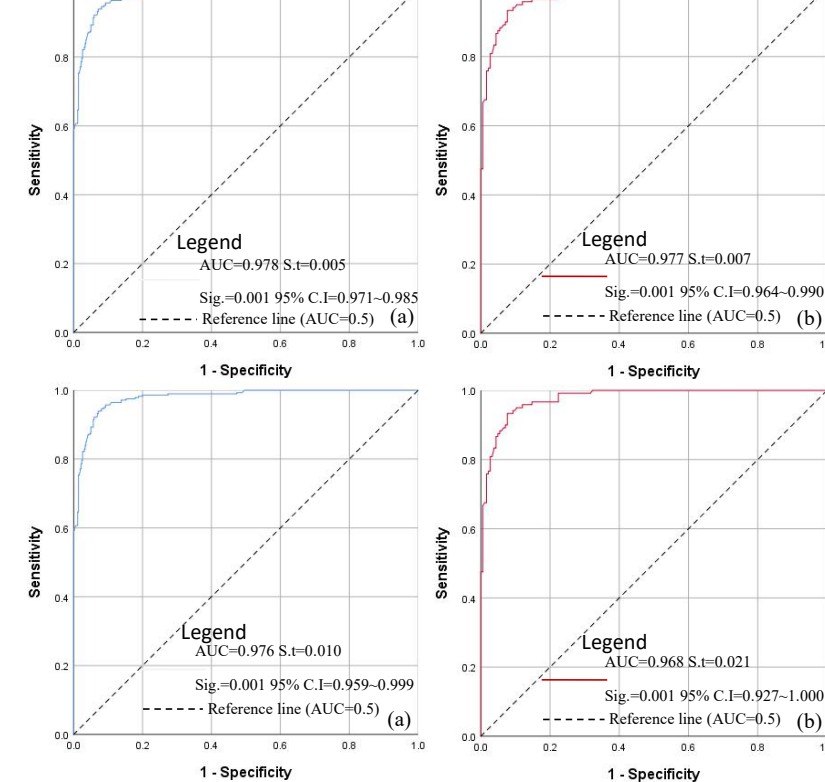

**Fig.7.** Analysis of ROC curve for the two susceptibility maps: **(a)** Success rate curve of landslide using
the training dataset; **(b)** Prediction rate curve of landslide using the validation dataset; **(c)** Success rate
curve of debris flow using the training dataset; **(d)** Prediction rate curve of debris flow using the
validation dataset.

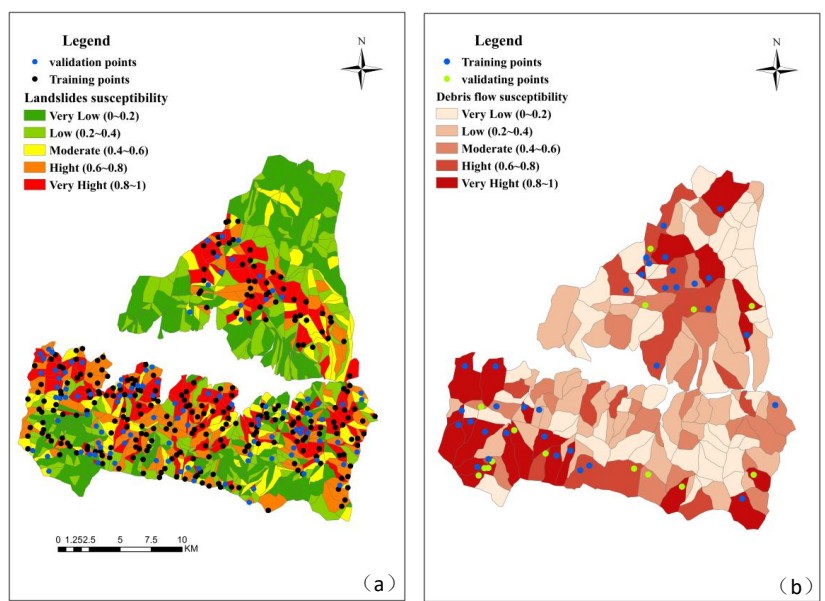

**Fig.8.** Susceptibility maps:(**a**)Landslide susceptibility zoning map;(**b**)Debris flow susceptibility zoing

map.

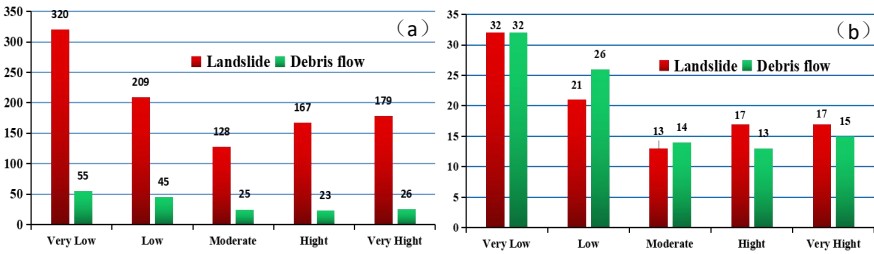

**Fig.9.** Numbers and percentage of units in different susceptibility classes for landslide and debris flow:

**(a)** Numbers of units in different susceptibility classes for landslide and debris flow; **(b)** Percentages of

different susceptibility classes for landslide and debris flow.


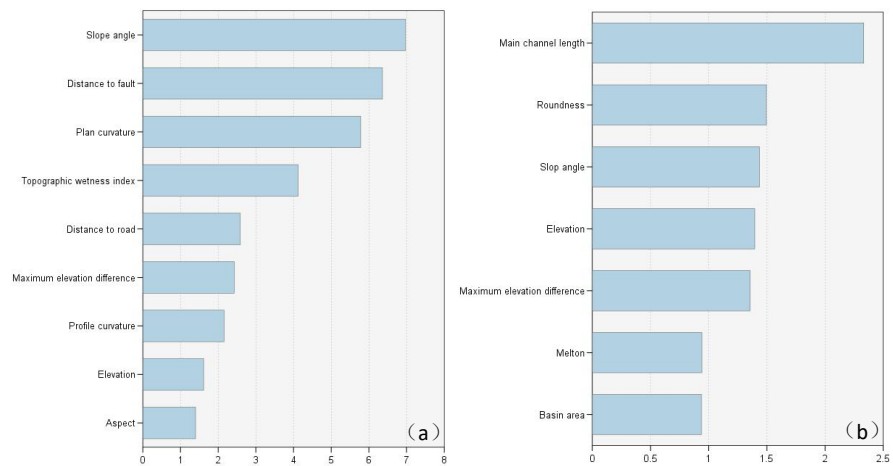


**Fig.10.** Parametric importance graphics obtained from RF model: **(a)** Parametric importance graphics
of landslide; **(b)** Parametric importance graphics of debris flow.



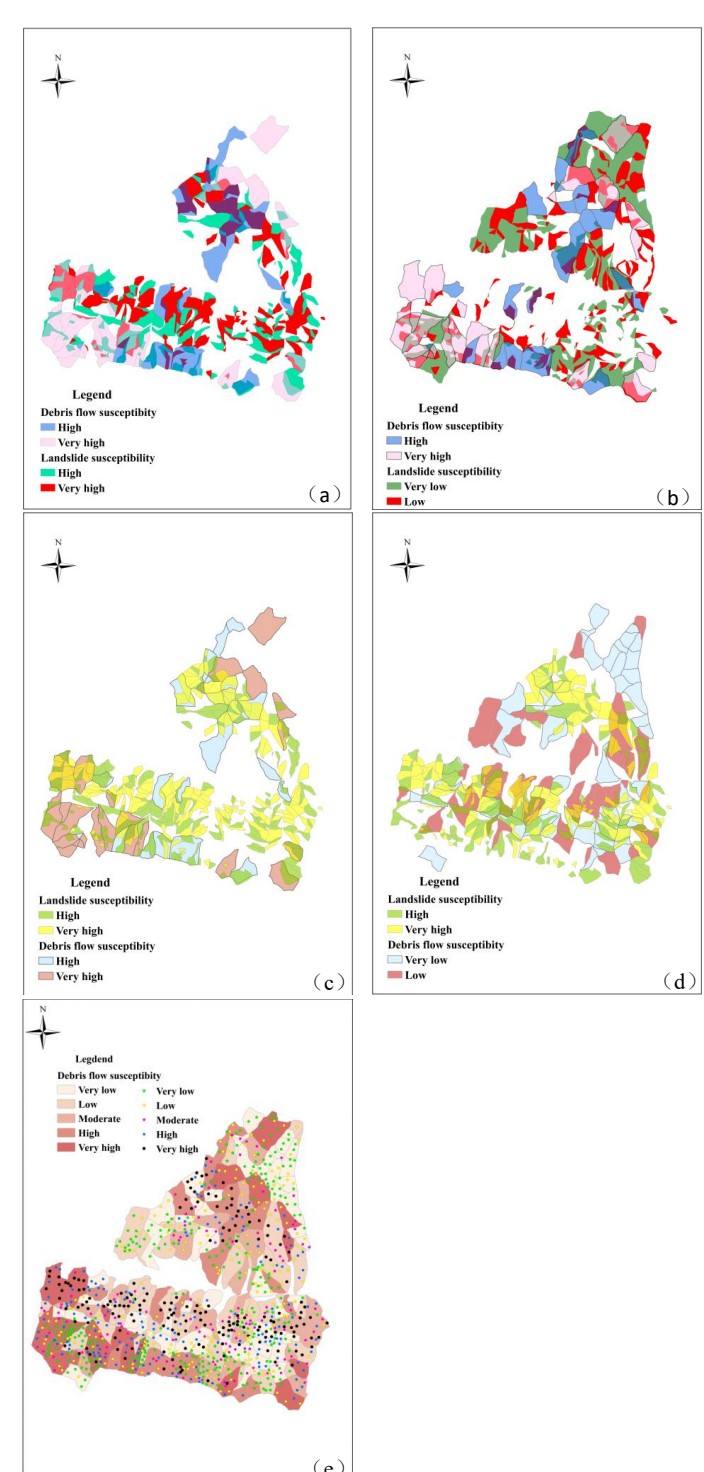





**Fig.11.** Landslide-debris flow susceptibility maps: **(a)** Height and very high-class watershed units with
high or very high slope units; **(b)** High or very high-class watershed units with low or very low slope
units; **(c)** High or very high-class slope units with high or very high-class watershed units; **(d)** Mapping
units.

