# Peer review of "Exploring the potential relationship between the occurrence"

_Natural Hazards and Earth System Sciences, 2020_

## Referee Comment (RC1) · Anonymous Referee #1 · 8 May 2020

The manuscript titled "Exploring the potential relationship between the occurrence of landslides and debris flows: A new approach" introduces a new way, which combines two susceptibility zoning maps to explored the potential relationship between two geological hazards (landslides and debris flows). The susceptibility assessment models of landslide and debris flow established by random forest have high accuracy and reliable results. Finally, the potential relationship between the two disasters was illustrated by superimposing the two maps, which also provides a basis for further research on the disaster chain. The manuscript is well-written, well-organized and innovative. The content involved in the article is very important in engineering prevention and control, and has certain reference value. For all these reason, I recommend the article for publica-

tion. However, the presentation of the manuscript can be improved after some minor revisions. 1. There is a problem with the number and order of the figures, and the figures do not correspond; 2. Figure 2 is composed of 5 pictures, but only 4 pictures description; 3. There is a problem with the format of Table 1, 2 and 5, please check; 4. Line 265, Fig8, it should be Fig.8 . 5. Why does Table 7 have 4 common factors and Table 6 not ? 6. The language used in the article needs further refinement.

---

## Author Comment (AC1) · 8 May 2020

Dear reviewer First of all, we would like to thank you for your careful work and valuable advice. We are very honored to receive your approval for our work. And now, we would like to answer your questions one by one. 1.We are sorry for the situation where the picture and text do not correspond. In fact, Figures 1 to 4 correspond to the first and second sections of Chapter 2. Figures 5 to 6 correspond to the 5 section of Chapter 2. Figures 7 to 11 correspond to the 4chapter; 2.There are some display errors in the a picture in fig.2, and we will correct this. In fact, fig. 2 consists of only 4 photos from the field survey; 3. We will correct them carefully; 4.We will correct them carefully; 5.In

application of factor analysis, if the cumulative contribution rate is greater than 85% or 80%, it is considered that most of the original information is retained. Therefore, the number of common factors may not be the same in different models; 6.We will further polish up the language. Thank you again for the valuable advice and it is very important for the improvement of our paper. best wish, Liang
* * *

---

## Referee Comment (RC2) · Anonymous Referee #2 · 9 May 2020

Dear authors,

I have revised the paper and decided to decline this paper for publication in NHESS with following reasons:

1. Methods used in this study is not new, this type of method and paper has a huge literature, the authors should develop more advanced techniques like hybrid or deep learning. 2. In my opinion, it is better to do seperate study for each type of landslides as different type of landslides has different mechanishm of occurrences, so, deep study is required. 3. What do you mean by landslide and debris flow. As I might know, debris flow is one type of landslides. 4. I have found a lot of writing and typing mistakes in this

verison. 5. Data sources are not given at all.

Best of luck!

---

## Referee Comment (RC3) · Anonymous Referee #1 · 9 May 2020

I'm satisfied with the authors' response.

---

## Short Comment (SC1) · 14 May 2020

I have read the MS entirely. The main content of this paper is to discuss the potential relationship between two common disaster in mountainous area. I think the new approach does not refer to random forests, but rather to the form of two disasters being discussed on the same susceptibility zoning map. From this point of view, the paper is innovative.The conclusion can also provide reference for the study of other forms of disaster chain.However, I found that the figures in the paper are in the wrong order, which is not correspond with the paragraph. Please double check.

2020-127, 2020.

---

## Author Comment (AC2) · 22 May 2020

Dear review First of all, thank you for your valuable time in reviewing this paper. And I am now I will reply to your comments one by one. 1.Random forest is a popular and efficient algorithm among many ensemble learning machines. Other ensemble machines, such as GBDT, AdaBoost-decsion Tree, etc., have also been apply to landslide susceptibility prediction. There are three basic ways of machine integration, Boostting, Bagging and Stacking. In addition, some deep learning machines, such as ANN and SVM, are also common in landslide sensitivity prediction. Of course, traditional modeling methods, such as logistic regression and bayes, have also achieved good

performance. The evaluation of a model is inseparable from several points: accuracy, robustness and analytical. Scholars have published a lot of papers on the comparison and application of evaluation methods in landslide susceptibility prediction, so the focus of this paper is not on the applicability of evaluation methods. But the application of random forest have achieve great performance, which guarantees the following research.The new approach referred to in the title of this article is not random forest, which may be a misunderstanding. 2.I have made relevant explanations of the point that different type of landslides has different mechanishm of occurrences. But, actually, there is some papers that do not differentiate between different types of landslides. I agree with your comment, so I selected different influencing factors for different types of landslides. 3. As you said, different type of landslides.Many studies show that landslides can be the source of debris flow. 4.We am sorry for the mistakes and please point them out. 5.It can be given at anytime if you need. Thank you again for the corrections you made. Best wish, Liang Relevant references: Guzzetti, F., Carrara, A., Cardinali, M., Reichenbach, P., 1999. Landslide hazard evaluation: a review of current techniques and their application in a multi-scale study, Central Italy. Geomorphology 31, 181–216. Guzzetti. F, M. Galli, P. Reichenbach, et al. Landslide hazard assessment in the Collazzone area, Umbria, Central Italy. 2006, 6(1):115. Guzzetti, F., Galli, M., Reichenbach, P., Ardizzone, F., Cardinali, M., 2006a. Landslide hazard assessment in the Collazzone area, Umbria, central Italy. Nat. Hazard. Earth Syst. Sci. 6, 115–131. Guzzetti, F., Reichenbach, P., Ardizzone, F., Cardinali, M., Galli, M., 2006b. Estimating the quality of landslide susceptibility models. Geomorphology 81, 166–184. Alessandro Trigila, Carla Iadanza, Carlo Esposito, et al. Comparison of Logistic Regression and Random Forests techniques for shallow landslide susceptibility assessment in Giampilieri (NE Sicily, Italy). 2015, 249:119-136. Ahmed Mohamed Youssef, Hamid Reza Pourghasemi, Zohre Sadat Pourtaghi, et al. Erratum to: Landslide susceptibility mapping using random forest, boosted regression tree, classification and regression tree, and general linear models and comparison of their performance at Wadi Tayyah Basin, Asir Region, Saudi Arabia. 2016, 13(5):1315-1318. Blais-Stevens A, Behnia

P (2016) Debris flow susceptibility mapping using a qualitative heuristic method and flow-R along the Yukon Alaska Highway Corridor, Canada. Nat Hazard Earth Syst Sci 16(2):449–462.

---

## Author Comment (AC3) · 22 May 2020

Dear Xiu Thank you for your recognition of our work.It is very ture that the focus of this paper is not on the evaluation method. After all, various types of evaluation methods have been widely used in landslide or debris flow susceptibility assessment. Therefore, our focus is to explore the potential relationship between two natural disasters through an excellent performance evaluation method. We are sorry for the mistake. We agree to recheck the serial number of the figures. Thank you again. Best wish, Zhu

---

## Author Comment (AC4) · 23 Jun 2020

It is our pleasure to have your recognition.

---

## Author Comment (AC5) · 23 Jun 2020

Dear review First of all, thank you for your valuable time in reviewing this paper. And I am now I will reply to your comments one by one. 1.Random forest is a popular and efficient algorithm among many ensemble learning machines. Other ensemble machines, such as GBDT, AdaBoost-decsion Tree, etc., have also been apply to landslide susceptibility prediction. There are three basic ways of machine integration, Boosting, Bagging and Stacking. In addition, some deep learning machines, such as ANN and SVM, are also common in landslide sensitivity prediction. Of course, traditional modeling methods, such as logistic regression and bayes, have also achieved good

performance. The evaluation of a model is inseparable from several points: accuracy, robustness and analytical. Scholars have published a lot of papers on the comparison and application of evaluation methods in landslide susceptibility prediction, so the focus of this paper is not on the applicability of evaluation methods. But the application of random forest have achieve great performance, which guarantees the following research. The new approach referred to in the title of this article is not random forest, which may be a misunderstanding. A new approach introduces a new way, which combines two susceptibility zoning maps to explored the potential relationship between two geological hazards (landslides and debris flows). 2.I have made relevant explanations of the point that different type of landslides has different mechanishm of occurrences. But, actually, there is some papers that do not differentiate between different types of landslides. I agree with your comment, so I selected different controlling factors for different types of landslides. 3.We have referred to relevant literature, and the classification of landslides is complex and diverse. The debris flow referred to in this paper refers to wet flow caused by rainfall, while landslide refers to various forms of rock or soil sliding that can provide a material basis for the occurrence of debris flow. Many studies show that landslides can be the source of debris flow. 4.We are sorry for the mistakes and please point them out. 5.It can be given at anytime if you need. Thank you again for the corrections you made. The focus of this paper is to explore the potential relationship between two geological hazards in a new way. What we want to emphasize is "way" rather than "method". Because there are too many papers on landslide susceptibility mapping, it is difficult to make new breakthroughs in both methods and mapping units. Therefore, our focus is on the relationship between the two types of landslides, intended to provide a new way to explore the disaster chain. As for the language, we will revise and polish it. Best wish, Liang

Relevant references: Guzzetti, F., Carrara, A., Cardinali, M., Reichenbach, P., 1999. Landslide hazard evaluation: a review of current techniques and their application in a multi-scale study, Central Italy. Geomorphology 31, 181–216. Guzzetti. F, M. Galli, P. Reichenbach, et al. Landslide hazard assessment in the Collazzone area, Umbria,

Central Italy. 2006, 6(1):115.

Varnes DJ (1978) Slope movement types and processes, in Schuster, R.L., and Krizek, R.J., eds., Landslides: Analysis and control, National Research Council, Washington, D.C., Transportation Research Board, National Academy Press, Special Report 176, p. 11–33

Guzzetti, F., Galli, M., Reichenbach, P., Ardizzone, F., Cardinali, M., 2006a. Landslide hazard assessment in the Collazzone area, Umbria, central Italy. Nat. Hazard. Earth Syst. Sci. 6, 115–131.

Guzzetti, F., Reichenbach, P., Ardizzone, F., Cardinali, M., Galli, M., 2006b. Estimating the quality of landslide susceptibility models. Geomorphology 81, 166–184.

Alessandro Trigila, Carla Iadanza, Carlo Esposito, et al. Comparison of Logistic Regression and Random Forests techniques for shallow landslide susceptibility assessment in Giampilieri (NE Sicily, Italy). 2015, 249:119-136.

Ahmed Mohamed Youssef, Hamid Reza Pourghasemi, Zohre Sadat Pourtaghi, et al. Erratum to: Landslide susceptibility mapping using random forest, boosted regression tree, classification and regression tree, and general linear models and comparison of their performance at Wadi Tayyah

Basin, Asir Region, Saudi Arabia. 2016, 13(5):1315-1318. Blais-Stevens A, Behnia P (2016) Debris flow susceptibility mapping using a qualitative heuristic method and flow-R along the Yukon Alaska Highway Corridor, Canada. Nat Hazard Earth Syst Sci 16(2):449–462.